# Cationic Magnetite Nanoparticles for Increasing siRNA Hybridization Rates

**DOI:** 10.3390/nano10061018

**Published:** 2020-05-27

**Authors:** Artur Y. Prilepskii, Arseniy Y. Kalnin, Anna F. Fakhardo, Elizaveta I. Anastasova, Daria D. Nedorezova, Grigorii A. Antonov, Vladimir V. Vinogradov

**Affiliations:** International Institute “Solution Chemistry of Advanced Materials and Technology”, ITMO University, 197101 St. Petersburg, Russia; prilepskii@scamt-itmo.ru (A.Y.P.); kalnin@scamt-itmo.ru (A.Y.K.); fakhardo@scamt-itmo.ru (A.F.F.); anastasova@scamt-itmo.ru (E.I.A.); nedorezova@scamt-itmo.ru (D.D.N.); antonov@scamt-itmo.ru (G.A.A.)

**Keywords:** magnetite, gold, siRNA, hybridization, nanoparticles

## Abstract

An investigation of the interaction principles of nucleic acids and nanoparticles is a priority for the development of theoretical and methodological approaches to creating bionanocomposite structures, which determines the area and boundaries of biomedical use of developed nanoscale devices. «Nucleic acid—magnetic nanoparticle» type constructs are being developed to carry out the highly efficient detection of pathogens, create express systems for genotyping and sequencing, and detect siRNA. However, the data available on the impact of nanoparticles on the behavior of siRNA are insufficient. In this work, using nanoparticles of two classical oxides of inorganic chemistry (magnetite (Fe_3_O_4_) and silica (SiO_2_) nanoparticles), and widely used gold nanoparticles, we show their effect on the rate of siRNA hybridization. It has been determined that magnetite nanoparticles with a positive charge on the surface increase the rate of siRNA hybridization, while negatively charged magnetite and silica nanoparticles, or positively charged gold nanoparticles, do not affect hybridization rates (HR).

## 1. Introduction

The history of studying the interaction between nucleic acids and nanoparticles (NPs) has been around for many decades. The focus of works on this subject has continuously shifted from topic to topic, covering such fields as biosensing, nanoassembly, targeted delivery, and gene therapy [1,2,3]. However, limited studies were directly dedicated to assessing the impact of nanoparticles of different materials on RNA hybridization. More specifically, a tempting question is the degree of the hybridization of double-stranded siRNA after interaction with inorganic nanoparticles.

Nowadays, the study of the effect of various nanoparticles on the hybridization of RNA or DNA structures seems to be very important. First of all, metals can coordinate hydrogen bonds between two strands [4] and can also replace those bonds, thereby making DNA or RNA structures more stable and developing new structures [5]. In this case, metal nanoparticles perhaps can also stabilize hydrogen bonds in DNA and RNA, so small amounts of nanoparticles can be used for the assembling of synthetic structures, such as DNA machines and siRNA. The second reason for this study is the development of delivery systems for RNA and DNA structures. Delivery is a common problem for all therapeutic agents that are based on nucleic acids [6]. It is well known that nanoparticles can easily penetrate cells [7,8,9,10] and also deliver nucleic acids [11,12].

Pioneering work of Chad Mirkin group on assembling nanoparticle aggregates through DNA has opened up significant prospects for the possibility of using natural molecules as guiding agents to create structures that are inaccessible to conventional chemical synthesis [13].

W.H. Kong and authors showed the formation of the Au-siRNA complexes while using Au-thiol linkages [14]. Another method, biosensing, is also based on the affinity of DNA to gold, but the principle of complementarity plays an important role here [15]. It allows for determining similarities between single-stranded DNA molecules, thereby revealing mutations or the presence of pathogenic DNA, while nanoparticles serve as an optical sensor. There are works where the reverse principle is used: after the self-hybridization of DNA on the surface of nanoparticles, the addition of any cleaving agents can destroy the aggregate and change the optical properties [16]. Magnetite and gold-coated magnetite nanoparticles were also used for DNA hybridization while using the streptavidin-biotin bridge [17].

In all of the works mentioned above, the formation of complexes siRNA or DNA with nanoparticles was reached using linkers. In this work, we found it interesting to explore the effect of different pristine nanoparticles without any coating and linkages on the rate of hybridization of siRNA sequences. Moreover, we investigated the possibility of the use of nanoparticles as a hybridization platform instead of a conventional melting-cooling technique. Magnetite, gold, and silica nanoparticles (MNPs, GNPs, and SNPs, respectively) were chosen for this purpose, as these systems were already used for siRNA or DNA hybridization [18,19,20]. The primary focus of experiments was directed to studying the HR of the same sense and antisense sequences via their addition and incubation into NPs solutions for different periods. It was shown that positively charged MNPs could increase the HR of siRNA molecules, while gold, silica, and negatively charged MNPs do not have a significant influence on the hybridization process.

## 2. Materials and Methods

### 2.1. Chemicals

Iron (II) chloride tetrahydrate (≥98.5%), iron (III) chloride hexahydrate (≥99%), sodium citrate, PEG 8k, gold (III) chloride solution (99.99%, 30 wt.% in dilute HCl), sodium borohydride (98%, powder), cetyltrimethylammonium bromide (99.0%, BioUltra, for molecular biology), SYBR Gold, acrylamide, bisacrylamide, urea, tris base, boric acid, ethylenediaminetetraacetic (disodium salt), tetramethylethylenediamine (TEMED) ammonium persulfate, and PBS were purchased from Sigma–Aldrich. Deionized water from Elix Essential 3UV, Millipore. Sodium silicate, acetic acid, acridine orange was obtained from Chimmed (Moscow, Russia). The oligonucleotides used in the study were ordered in the IDT (Coralville, IA, USA), dissolved in RNAse/DNAse free water to final concentration 100 μM, pH 5.5, and were stored frozen. Before the experiments, the stocks were dissolved to 10 μM in RNAse/DNAse free water.

Oligonucleotides sequences used in the study: DAD1_sense CCACACCGCAGCGUCUGAAUU; DAD1_antisense UUCAGACGCUGCGGUGUGGGA.

### 2.2. Synthesis of NPs Sol

The scheme of synthesis is provided in Appendix A. In brief, magnetite hydrosol was prepared in ultrasonically assisted synthesis from iron (II) chloride tetrahydrate and iron (III) chloride hexahydrate, as described previously [21]. The mass fraction of magnetite nanoparticles with average particle size 10 nm in the resulting sol was 2 wt.%. To prepare negatively charged MNPs^−^, 1 mL of the fresh sol was mixed with 100 µL 0.36 M sodium citrate and then incubated overnight on an orbital shaker. Pegylated MNPs^P^ were prepared by mixing 100 µL of 100 mM PEG with 1 mL of fresh MNPs, followed by overnight incubation on an orbital shaker. Gold nanoparticles with different sizes (GNPs-15 and GNPs-30 with an average diameter of 15 nm and 30 nm, correspondingly) were synthesized from chloroauric acid and an ice-cooled solution of sodium borohydride, which were added to the CTAB solution under vigorous stirring on a magnetic stirrer [22]. The mixture was washed with deionized water to neutral pH and treated in an ultrasonic bath for 1 h to stabilize the system. The resulting GNPs have octahedral form and size of 30 nm. Silica nanoparticles (SNPs) were prepared by mixing 3 mL of acetic acid with 50 mL of deionized water and the subsequent addition of 2 mL of 25% sodium silicate. All of the nanoparticles were purified by centrifugation. Concentrations of nanoparticles in terms of particles/mL were calculated theoretically. The mass of the initial precursors was divided by the density of the material. The resulting whole volume of nanoparticles was divided by the average volume of single nanoparticle, assuming that nanoparticles were spherical. 

### 2.3. Characterization Techniques

The crystal phase and crystallinity of the samples were studied by the X-ray diffraction method (Rigaku SmartLab 3 diffractometer of the Engineering center of the Saint-Petersburg State Technological Institute (Technical University)) using Cu-Kα irradiation (λ = 1.54 Å), with samples being scanned along 2θ in the range of 5–80° at a speed of 0.5°/min. For XRD analysis, the samples were dried at 120 °C for 4 h. The particle size and zeta potential of NPs were measured using a Photocor Compact Z. For SEM analysis, the samples were dried in vacuo for 1 h and then examined using a Tescan VEGA 3 electron microscope. The FTIR spectra were obtained by Nicolet iS5 spectrometer. The samples were dried in oven at 60 °C overnight, powdered with agate mortar, and then embedded in mineral oil for measurement.

### 2.4. Fluorescent Measurements

Fluorescence measurements with acridine orange were performed, as follows. Three different concentrations of GNPs were prepared: the stock solution, the stock solution diluted 10 times, and the stock solution diluted 100 times. 1 µL of acridine orange (0.5 µg/mL) was mixed with different volumes of these three GNPs solutions, namely, 1, 2, and 5 µL (additional sample with 10 µL was made for stock solution). The whole volume of probes was brought to 100 µL with water. These probes were used as reference values. For probes with siRNA, 1 µL of S and As siRNA solutions (1 µM) were added. All of the probes were left at room temperature for 30 min., and fluorescence was measured on Cary Eclipse spectrofluorometer at the excitation wavelength of 490 nm and emission wavelength of 525 nm.

### 2.5. Nanoparticles-Assisted siRNA Hybridization

We conducted three experimental schemes to evaluate how nanoparticles can affect the rate of hybridization of siRNA strands. In the first scheme, NPs were added (1 µL) to the mixture (20 µL) of (S + As) (e.g., (S + As) + MNPs). In the second scheme, solutions of sense (S) (10 µL) and antisense (As) (10 µL) sequences were separately mixed with MNPs, SNPs, or GNPs (0.5 µL), followed by the mixing of prepared solutions (e.g., (S + MNPs) + (As + MNPs)). In the third scheme, only siRNA sequences were mixed (20 µL) to examine the positive or negative action of NPs (this scheme used as control). Each tube contained 6 × 10^12^ of sense and antisense siRNA molecules, 6 × 10^8^ NPs (first and second schemes), so the siRNA molecules were in excess, and distilled water, with total volume approximately 20 µL. After mixing, each tube was placed to a constant temperature of 23 °C and then incubated for fixed periods (0, 0.5, 1, 2, or 3 h), with the subsequent addition of 100 mM PBS (pH = 7.4) (4 µL), for sedimentation NPs, and centrifugation at 14,000× *g* for 5 min. 

### 2.6. Evaluation of RNA Hybridization Level Using PAGE

The supernatant obtained from centrifugation was analyzed using native polyacrylamide gel electrophoresis (PAGE). The Mini-PROTEAN^®^ Tetra system (BioRad, Hercules, CA, USA) was used for PAGE. The polyacrylamide gel was obtained by mixing 7 M Urea, 40% Acrylamide-Bisacrylamide solution, 10X TBE buffer (Tris base (0.9 M), Boric acid (0.9 M), EDTA (disodium salt) (25 mM), and deionized water. Tetramethylethylenediamine (TEMED) and 10% ammonium persulfate solution (APS) were used for gel solidification, followed by pouring the gel between two foresis glasses and comb was used for sample holes formation. After that, the solidification comb was taken out of the gel, and gel in glasses was placed in a gel cassette, which was placed in the electrophoresis tank. 1x TBE buffer was poured in both gel cassette and electrophoresis tank. 10 µL of the sample was mixed with 2 µL of loading buffer and then added into the gel. The first five lanes in each gel were experimental samples from different incubation times and with nanoparticles, and the last five lanes were used for control (same incubation time but without nanoparticles). PAGE conditions: 80 V, 120 min., 20% gel density. Immediately after the end of electrophoresis, the gels were dyed with SYBR Gold according to the kit protocol and then visualized. 

### 2.7. Analysis of Gel Images

Gel pictures were analyzed, as follows. Each band was saved as a separate image of the same size in pixels, and the total intensity of pixels was calculated using Wolfram Mathematica software with in-build function ImageData. The hybridization rate (${\mathrm{HR}}_{t}$) of siRNA at the specified time *t* was calculated as a ratio of the value of mean upper band intensity (hybridized siRNA) to mean total intensity (hybridized and non-hybridized siRNA) and expressed in percentage:(1)$${\mathrm{HR}}_{t}=100\times \frac{\frac{1}{n}\times \sum_{1}^{n}A_{n}^{t}}{\frac{1}{n}\times \sum_{1}^{n}\left(A_{n}^{t}+B_{n}^{t}\right)}$$
where $A_{n}^{t}\;$—the brightness of nth upper band (hybridized siRNA) at time *t*, and $B_{n}^{t}$—the brightness of the nth lower band (non-hybridized siRNA) at time *t*. $\frac{1}{n}\times \overset{n}{\underset{1}{\sum }}A_{n}^{t}$ mean intensity of *n* upper bands and $\frac{1}{n}\times \overset{n}{\underset{1}{\sum }}\left(A_{n}^{t}+B_{n}^{t}\right)\;$—mean value of a sum of upper and lower band intensities. For all experiments, *n* = 5. An example of calculation along with raw data are presented in Appendix A.

### 2.8. Statistical Analysis

Data were processed using conventional methods of variation statistics. Differences between groups were considered to be significant based on Student’s *t*-test.

## 3. Results and Discussions

The hybridization process of single-stranded nucleic acids is a well-known process of bond formation between complementary individual bases according to Chargaff’s rules. Despite the apparent simplicity of the process, at least four main forces are involved in the process of hybridization and the further stabilization of double helix: (i) hydrogen bonds between base pairs; (ii) hydrogen bonds between base pairs and surrounding water molecules; (iii) stacking interactions between neighbor base pairs; and, (iv) repelling forces between backbone phosphate groups [23]. All of these forces must be taken into account when discussing the precise mechanism of strands interaction. Natively present siRNA is much more complicated than used in this work and it typically has 50–300 base pairs, loop regions, and single-stranded regions, i.e., possessing some self-assemble features [24]. In solution, complementary strands can partially hybridize, and a standard melting/cooling procedure is employed to enhance the percentage of hybridized RNA/DNA and the quality of this self-hybridization. When the double-stranded nucleic acid is heated, the hydrogen bonds break, and molecules separate into two single strands during a process that is known as nucleic acids melting. When a solution is cooled, complementary bases realign and bind through hydrogen bonding. However, it initially needs to overcome repulsive forces between phosphate groups, while many divalent ions (such as Mg^2+^) decrease these forces, thus helping the hybridization process [25]. Subsequently, hydrogen forces bonds come into play, but they are too weak to hold two strands together in a case when only some of the base pairs meet their complementary ones [26]. Eventually, after a number of these interactions, two strands complementary bond to each other and form a helix. However, a simple progression of helix formation can be problematic due to accumulated torsional stress in the strands, and spatial freedom is essential in this step [27]. Several pieces of research on the topic of siRNA hybridization give some characteristics of a typical pairing rate, which is 10^6^ M^−1^ s^−1^ [27]. To sum up, hybridization is a multi-step process that involves not only «chemical», but also mechanical forces, and it could be significantly altered by various additives. Thus, we addressed three questions in this work. (I) Can nanoparticles act as a hybridization platform instead of a conventional melting-cooling technique? (II) What the effect of nanoparticle parameters (size, surface area, charge, etc.) on siRNA hybridization? (III) What is the possible mechanism of siRNA hybridization on the surface of NPs?

To answer these questions, we precisely investigated the physicochemical properties of the nanoparticles. According to the TEM images (Figure 1), the synthesized materials have different morphologies (MNPs and SNPs—spherical, GNPs—octahedral) (Figure 1), wherein the particle size distribution according to SEM was quite narrow (Appendix A, insets). The crystal structure of the MNPs and GNPs was investigated by XRD with a crystallite size of 10 and 28 nm, respectively (Figure 2A), while SNPs were completely amorphous (no peaks were observed). The X-ray pattern of MNPs corresponds to the magnetite phase (JCPDS file No. 19-0629) and the diffraction patterns for GNPs revealed the diffraction peaks for cubic gold (JCPDS file No. 2-1095). Table 1 shows the hydrodynamic diameter and ζ-potential of NPs in distilled water. The hydrodynamic diameters of particles ranged from 30 nm to 72 nm (Figure 2B). However, the actual diameter of nanoparticles ranged from 4 to 30 nm (according to SEM images, Appendix A). The difference arises from the principle of DLS measurement, which displays the size of the hydrated nanoparticle. In our further calculations, we used hydrodynamic diameter, because it better represents the actual behavior of nanoparticles in media. We used ζ-potential-to-hydrodynamic diameter ratio in our further consideration instead of ζ-potential-to-actual size since ζ-potential originates from DLS.

MNPs and GNPs both have positively charged surfaces, while SNPs were negatively charged. The surface of MNPs was additionally investigated via infrared attenuated total reflection (IR-ATR). Spectra were recorded in a sequence of temperatures (25–200 °C) on the same sample by heating it gradually. The dominating peaks on the IR-ATR spectra of MNPs powder are presented in Appendix A and they correspond to water bounded to the surface of MNPs. The sample was gradually dehydrated directly on the ATR stage until reaching and stabilizing the temperature and, afterward, the spectra at that temperature were recorded in order to characterize the surface of particles. Even at the temperature of 25 °C, the OH stretching vibrations of the surface OH groups are detectable with a wavelength around 3749 cm^−1^, which can be attributed to the surface OH groups in magnetite/maghemite according to the literature data [28]. The very broad and intense band with minimum ATR intensity at approximately 3344 cm^−1^ is changing in intensity, so that this band is absent in the spectrum recorded at 200 °C. It can be attributed to incorporate both the symmetric and asymmetric stretching vibrations of the molecular water adsorbed on the sample surface. The diminishing of this band with the rise of the temperature reveals the overlapped band with a minimum in the range 3605–3625 cm^−1^ (due to the small signal/noise ratio in this region). It is a complex and relatively broad band with a low intensity that originates from the presence of the Fe(OH)_2_ on the surface of the investigated particles [28,29].

The properties of metal sols (for example, their activity and selectivity as catalysts for various chemical reactions) strongly depend on the size, shape, and charge of nanoparticles. That is why we also altered surface charge and surface coating of MNPs. The MNPs were recharged with sodium citrate to form negatively charged MNPs (MNPs^−^) and coated with PEG to form pegylated MNPs (MNPs^P^). FTIR spectra were measured to prove the coating. We observed a shift from 1700 cm^−1^ to 1600 cm^−1^ in MNPs^−^ sample, attributed to the functionalization of MNPs surface via carboxyl groups of sodium citrate (Appendix A). As for MNPs^P^, we observed a summary peak at 1100 cm^−1^ for C–O stretch of PEG molecule.

As a result, we obtained a broad range of different combinations of charge-to-size ratios (presented in Table 1). We also calculated the ratio of ζ-potential to the NPs size. SNPs have the lowest negative charge, but smallest diameter, possessing −0.33 a.u. (here and further are mean values presented). MNPs^−^ have a twice higher value of −0.63 a.u. MNPs and MNPs^P^ both have almost the same values of +0.55 and +0.53 a.u. respectively, but MNPs^P^ has a PEG shell on the surface, which can somehow affect the hybridization. GNPs-30 have the lowest positive charge and value of +0.33 a.u., while GNPs-15 has the highest value of 1.18. Also, according to SEM data, GNPs-30 have an average diameter of 30 nm, which is more than four times bigger than of MNPs (though the hydrodynamic size of these nanoparticles is almost identical).

We mixed the siRNA strands with NPs to evaluate the influence of NPs on the rate of siRNA hybridization and stability of complexes in time, as described in the Materials and methods section, following by separation in polyacrylamide gel electrophoresis (PAGE). Before this, we evaluated an approximate number of siRNA molecules that can bond to the NPs surface. For this purpose, we took GNPs-30 and stained the siRNA solution with acridine orange (intercalating fluorescent dye). GNPs was chosen because it was the only sol among our samples that known for its ability to quench acridine orange fluorescence in stained nucleic acids upon their binding to GNPs surface [30]. Approximation shows that zero fluorescence signal intensity can be reached at a 5000:1 ratio between the number of siRNA molecules and gold NPs (Figure 3). That value should correspond to the saturation of GNPs surface, with all siRNA molecules being attached. Based on this, we take siRNA in some excess and maintain constant ratio siRNA:NPs in all samples. Sense and antisense sequences were added to the NPs solution in two different ways. First scheme: NPs were added to premixed S and As sequences ((S + As) + NPs). Second scheme: sense and antisense strands were separately mixed with NPs, followed by mixing of (S + NPs) and (As + NPs) complexes together. The same schemes of mixing, but without NPs, were used as control.

Figure 4B presents an example of PAGE. Hybridized siRNA migrates in gel slower than single-stranded sense (S) and antisense (As) siRNAs because it has higher molecular weight, than separate siRNA strands. So, the upper bend in gel corresponds to the hybridized siRNA, while the lower bend is single-stranded sense (S) and antisense (As) siRNAs. The intensity of bands (in terms of greyscale) corresponds to the amount of siRNA in the band. Particularly, in Figure 4B, lanes 1 to 5 is a sample with MNPs and lanes 6–10 is a control sample without nanoparticles. As we can see by a naked eye, the intensity of upper bands in lanes 1–5 much higher than of upper bands in lanes 6–10. To prove it quantitatively, we performed image analysis as described in the Methods section.

The steadily higher HR was detected only for the first experimental scheme using MNPs (Figure 4A). Immediately after mixing of (S + As) with MNPs, an 18% increase in the hybridization value was observed. However, we observed vast fluctuations of hybridized siRNA concentration during further incubation, which can be attributed to the heterogeneity of the system, and the formation of unstable aggregates (Appendix A). Over the next two hours, the system becomes more stable, the hybridization rate of the control sample gets 5–10% faster, and the system reaches the plateau with the final HR value of MNPs sample statistically indistinguishable from the control sample (Appendix A). The overall kinetics of the hybridization reaction in the sample with MNPs suggests that the total amount of hybridized siRNA remains almost the same during the entire 3 h of the experiment, while the control sample shows a steady increase in HR. This results in a decrease in the difference in HR values between experimental and control samples after 3 h. In contrast to MNPs, we observed no statistical difference in HR for GNPs-30 and SNPs at any time of incubation (Appendix A). Surprisingly, we were not able to detect any siRNA in the sample that was incubated with GNPs-15. We can suggest that this is connected with high surface charge of nanoparticles that not allowing desorption of siRNA via the addition of PBS (Appendix A).

We want to propose a hypothesis to explain the observed phenomenon of siRNA hybridization on the surface of NPs (Figure 5). We are assuming that the primary reason behind higher hybridization rates of siRNA in the presence of MNPs is related to the binding of siRNA strands to the surface of MNPs or with the locally increased concentration of siRNA in MNPs surrounding. This increases the chances of siRNA strands to meet and bond with each other.

Surface charge plays a crucial role in the process of siRNA binding to the surface of MNPs. siRNA is mostly negatively charged due to phosphate groups and it readily binds to positively charged MNPs. However, we observed a considerable difference in hybridization rates between samples with MNPs and GNPs, which are both positively charged. The possible mechanism of better siRNA hybridization on the surface of MNPs is possibly connected with the optimal surface charge to nanoparticles size ratio. Having almost the same ζ-potential, MNPs and GNPs differ four times in diameter. When considering the length of the siRNA molecule, which is about 7.5 nm [31], we can suggest, that upon binding to bigger GNPs (d = 30 nm), siRNA changes dimensional conformation and lose spatial mobility which is crucial for helix formation [27]. On the other hand, MNPs with an average diameter of 7 nm probably bind siRNA strands only partially, leaving free ends for hybridization. Figure 5 schematically presents this suggestion. One of our first suggestions to explain the increase of HR was based only on the impact of the surface area of nanoparticles. However, for small GNPs-15 surface area was the same, but we observed no increase in HR. We proposed that the charge of NPs was too big to desorb siRNA since we were using PBS as a phosphate groups donor to desorb siRNA from the surface of nanoparticles. This allows for us to conclude that charge-to-size ratio should be in an optimal range for increasing HR.

We performed additional experiments with negatively charged MNPs and PEG-coated MNPs^P^ to verify the hypothesis that the HR depends mostly on charge-to-size ratio and charge sign. Figure 6 presents the results. We observed no increase in HR with MNPs^−^, which indicates that the composition of NPs (e.g., [Fe^2+^] and [Fe^3+^] ions) does not play a key role in the hybridization process. Concurrently, MNPs^P^ shows an almost 8% increase in HR. Taking into account the considerable broadening of hydrodynamic sizes of MNPs^P^ upon pegylation, we can speculate that there is some critical value of the charge-to-size ratio that assists hybridization. Additionally, surface modification and its effect on HR needs further investigation. The highest value of charge density for MNPs^P^ is 0.55 a.u., which is almost the same as for pure MNPs, as can be seen from Table 1. According to our assumption, high charge density is beneficial, but not in this case, since HR for MNPs^P^ is only 8% higher than the control value. Additionally, MNPs have a relatively narrow range of ratio values, which means that almost all of the particles can act as effective centers of hybridization, while in other systems particles meet these criteria only partially.

## 4. Conclusions

Summarizing presented results, we can conclude that composition, size, and surface parameters (namely, ζ-potential), and especially their ratio, should always be considered when the nucleic acid is meant to be used along with nanoparticles. Specifically, we want to emphasize a pronounced result of increased HR of siRNA in the presence of positively charged magnetite (Fe_3_O_4_) nanoparticles. We assume that having the optimal value of the charge-to-size ratio and pure surface, MNPs act as active centers for hybridization. On the other hand, GNPs-30 are too big to provide spatial mobility for helix formation, and GNPs-15 have a too high surface charge to allow for the desorbing of siRNA. Such a phenomenon can be further considered in the development of various delivery systems that are based on siRNA. For example, MNPs can be targeted to the desired site in the human body and enhance the hybridization rate of single siRNA strands on demand. This can be used in cancer treatment or other types of therapy.

## Figures and Tables

**Figure 1 nanomaterials-10-01018-f001:**
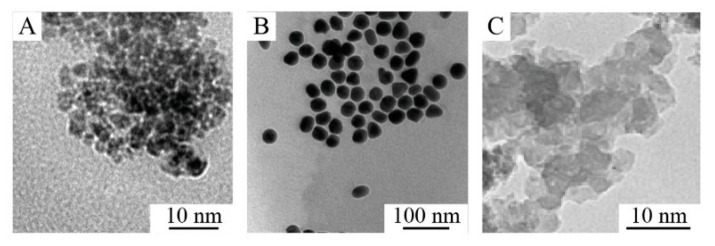
TEM images of nanoparticles (NPs). (**A**) Magnetite nanoparticles (MNPs). (**B**) GNPs-30. (**C**) Silica nanoparticles (SNPs).

**Figure 2 nanomaterials-10-01018-f002:**
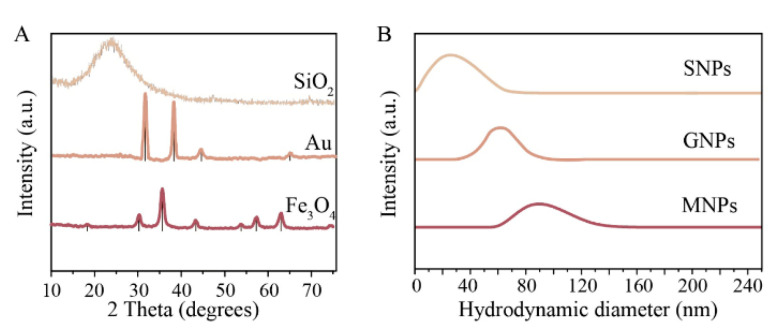
(**A**) XRD patterns of SNPs (**top**), GNPs (**middle**), and MNPs (**bottom**). Black vertical lines are showing reference to JCPDS file No. 19-0629 and JCPDS file No. 2-1095. (**B**) hydrodynamic diameter distribution of SNPs (**top**), GNPs-30 (**middle**), and MNPs (**bottom**).

**Figure 3 nanomaterials-10-01018-f003:**
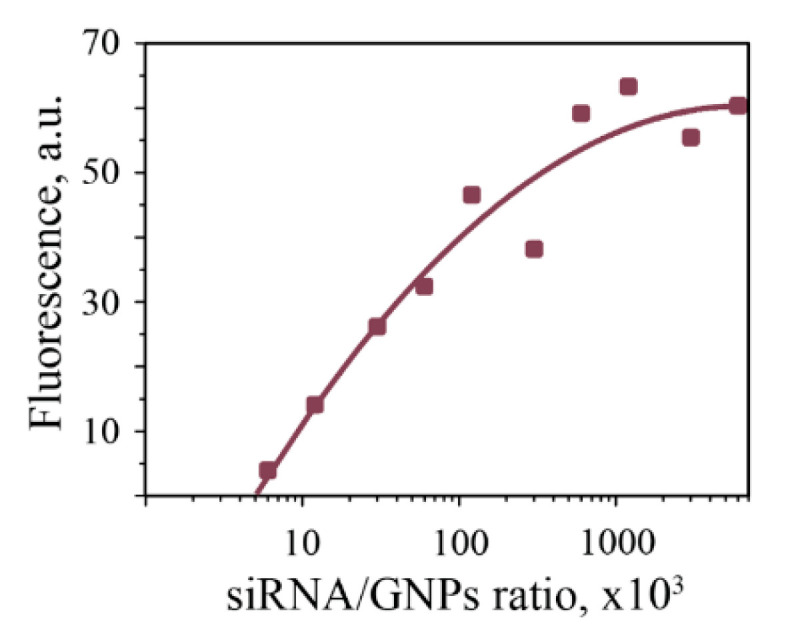
Relative fluorescence intensity of gold nanoparticles-siRNA (GNPs-siRNA) complexes, stained with acridine orange.

**Figure 4 nanomaterials-10-01018-f004:**
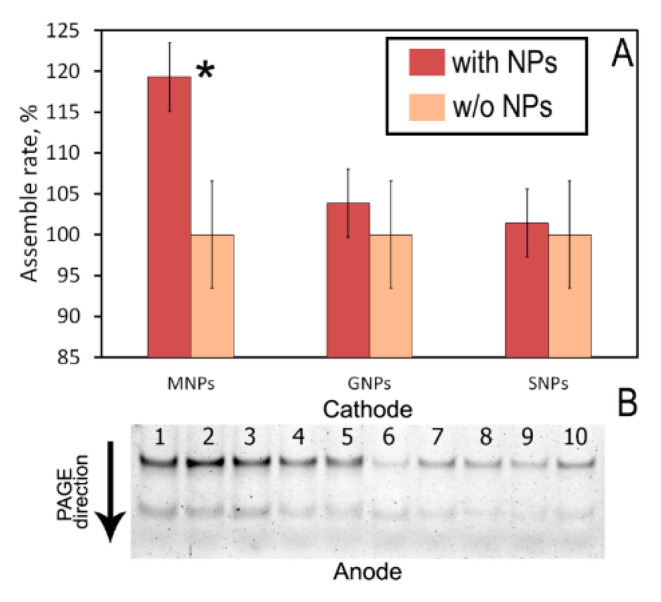
(**A**) Percentage of hybridized siRNA molecules in the presence of NPs. Shown values are an average of 5 repeats in relation to control immediately after mixing of siRNA with NPs. Error bars represent standard deviation. * *p* < 0.05. (**B**) Image of PAGE gel with (S + As) + MNPs (lanes 1–5) and (S + As) (used as the control, lanes 6–10) immediately after mixing. Upper and lower bands are hybridized and non-hybridized siRNA, respectively.

**Figure 5 nanomaterials-10-01018-f005:**
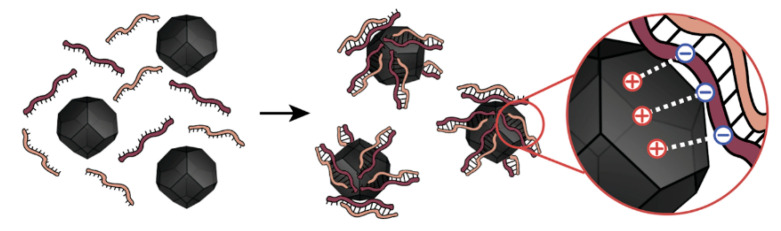
The proposed mechanism of siRNA hybridization on the surface of MNPs. Due to the electrostatic interaction between small positively charged MNPs and negatively charged siRNA strands, there is a local increase in the concentration of siRNA on the surface, leading to their hybridization.

**Figure 6 nanomaterials-10-01018-f006:**
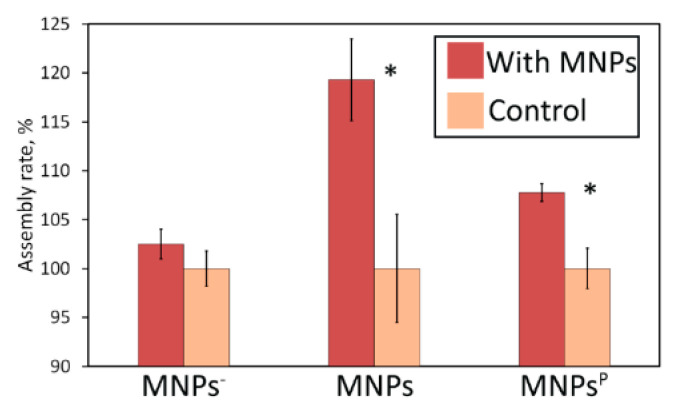
HR in the presence of MNPs, MNPs^−^, and MNPs^P^ immediately after mixing siRNA with NPs. * *p* < 0.05.

**Table 1 nanomaterials-10-01018-t001:** Hydrodynamic size, ζ-potential of NPs, and their ratios.

Sample	Hydrodynamic Diameter, nm	pH	ζ-Potential, mV	Range of ζ-Potential-to-Diameter Ratio * (Mean Value), a.u.
SNPs	30 ± 15	5.2	−10 ± 0.2	−0.68–0.22 (−0.33)
MNPs	60 ± 10	7.4	+33 ± 0.7	0.46–0.67 (0.55)
MNPs^−^	72 ± 16	8.0	−45 ± 0.6	−0.81–0.50 (−0.63)
MNPs^P^	62 ± 20	4.1	+33 ± 0.4	0.39–0.79 (0.53)
GNPs-30	60 ± 20	5.6	+20.0 ± 0.3	0.24–0.50 (0.33)
GNPs-15	30 ± 10	5.6	+35.4 ± 0.5	0.87–1.79 (1.18)

* Presented values are lowest and highest ones calculated taking into account standard deviation of sizes and ζ-potentials.

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
