# Peer review of "Cationic Magnetite Nanoparticles for Increasing siRNA Hybridization Rates"

_nanomaterials, 2020, doi:10.3390/nano10061018_

Round 1

Reviewer 1 Report

In this manuscript, Prilepskii and colleagues investigated the effect of nanoparticles on siRNA hybridization rate. The authors prepared a series of inorganic nanoparticles and tested their ability to increase siRNA hybridization rate. A mechanism was proposed stating that the siRNA hybridization rate could be modulated by the electrostatic interaction between siRNA backbone and nanoparticle surface. While this is an interesting study, there are still some critical questions and writing issues that need to be thoroughly addressed (see comments below). These issues necessitate a minor revision to this manuscript before it can be considered for acceptance.

Additional Comments:

  1. Figure 1 was referred to as “SEM images” in the text, where it should be “TEM images”.
  2. The size distribution of MNP and SNP in Table 1 is different from the particles shown in Figure 1 (<10 nm particles). What is the potential reason for this difference? Please explain.
  3. Given the different sizes of nanoparticles available, it would be interesting to see whether nanoparticle size (thus surface area) plays a role in the hybridization rate. Please discuss potential contribution of particle size or include some results on this direction.

Author Response

Response to Reviewers remarks

We want to thank Reviewers for their time, and constructive suggestions are given for the manuscript entitled «Cationic Magnetite Nanoparticles for Increasing siRNA Hybridization Rates», written by Prilepskii et al.

We have carefully evaluated the comments and recommendations of the referees and revised the manuscript accordingly. Please find below a detailed point-by-point reply to all reviewer’s comments. We sincerely hope that we have succeeded in clarifying and answering all the queries that were raised.

Reviewer: 1

1) Figure 1 was referred to as “SEM images” in the text, where it should be “TEM images”.

We are sorry for this mistake; the corresponding change was made on Page 4, line 191.

2) The size distribution of MNP and SNP in Table 1 is different from the particles shown in Figure 1 (<10 nm particles). What is the potential reason for this difference? Please explain.

            The data presented in Table 1 measured via the DLS technique. Presented size is the hydrodynamical size and, in general, consists of the core (nanoparticle) and surrounding (solvent/water). In the case of small nanoparticles, hydrodynamical size can be 2-3 times bigger than the actual size of nanoparticle on SEM or TEM images. We added an explanation on Page 5, lines 199-204.

3) Given the different sizes of nanoparticles available, it would be interesting to see whether nanoparticle size (thus surface area) plays a role in the hybridization rate. Please discuss the potential contribution of particle size or include some results on this direction.

            Indeed, it was our first thought that the bigger surface area of smaller nanoparticles (namely, magnetite) is beneficial for increasing of hybridization rate. However, gold nanoparticles with the same size according to TEM images (GNPs-15) and almost the same z-potential +35mV did not increase HR. Also, MNPsP differs from MNPs by only 10 nm in size, but this already leads to a decrease in HR. As we stated in conclusion, size certainly affects HR but only in conjunction with z-potential. We have added some additional discussion about this to the top of Page 9.

Reviewer: 2

1)  As stated in the introduction (L30), “… a tempting question is the quality of the hybridization of double-stranded siRNA after interaction with inorganic nanoparticles.” The authors do not really address this issue.

In this sentence, we wish to say about the amount of hybridized siRNA in comparison with non-hybridized. We changed the “quality” for “degree” to better represent our thoughts.

2) Gold nanoparticles are initially referred to as GNPs. Then GNPs-15 and GNPs-30 appear, without explanation. Please comment.

The number after abbreviation signifies the average diameter of the nanoparticles. We added an additional explanation of this on Page 2, Lines 86-87.

3) The size of NPs as determined by SEM, TEM, and DLS are not consistent (MNP : 7 to 60 nm ; SNP : 4 to 30 nm ; GNP : 30 to 60 nm). Please comment. Do the authors talk about nanoparticles or aggregates (as observed in Fig. 1 and fig. S1) ?

4) As part of the discussion is focused on the interpretation of the zeta potential/size ratio, could the authors indicate/rationalize why they used the size as determined by DLS rather than by TEM or SEM ?

We added an additional explanation on this issue on Page 5, lines 199-204. Briefly, we used hydrodynamic diameter from DLS because it is better representing the behaviour of nanoparticles (especially of non-spherical forms) in media. Also, since ζ-potential measured by DLS too, we considered it more correct to calculate the “ζ-potential-to-hydrodynamic diameter ratio” instead of “ζ-potential-to-actual size”. As for the aggregates of SEM and TEM images, it originates from sample preparation. When a drop of sol is dried under vacuum before the electron microscopy observation, it is inevitably shrinking, causing aggregation of nanoparticles. Our observation has shown that all of the sols used were sedimentary and aggregately stable during the experiments. So, the presented in Figure S1 distribution is for separate nanoparticles.

5) The nanoparticles described in the manuscript are likely titratable. The zeta value for such particles depends on pH. Thus, the pH value should be indicated in the text. Furthermore, the NP-RNA interaction is sensitive to pH. Consequently, this reviewer thinks that a proper comparison of the various NPs should be carried out in buffered solution, at a fixed pH.

We updated Section 2.5 on Page 3 with volume information of mixed reagents. For our experiments, we used very small amounts of NPs (0.5 to 1 µL, diluted ~100 times from stock solution) in comparison with buffered siRNA (20 µL). In this case, pH values for all samples were the same, around 5.5. Nevertheless, we added the pH values of NPs to Table 1 on Page 6.

6) Complementary analytical data for citrate- and PEG-loaded MNPs should be provided to assess their decoration/coverage by citrate and PEG moieties (FT-IR, elemental analysis…). In addition, it is unclear how the various NPs were purified. A description should be provided.

            All of the nanoparticles were purified by centrifugation according to original procedures described in references 21 and 22. We added this statement to Page 2, lines 92-93. We also added FTIR spectra for MNPs- and MNPsP to Supporting Information (Figures S3 and S4), and a short explanation to the manuscript on Page 6, as well as to Materials and Methods section on Page 3, lines 104-106.

7) NP-assisted siRNA hybridization: Please indicate how was calculated/measured the number of NPs (6 X 108) in the samples.

            Concentrations of nanoparticles (particles/mL) were calculated theoretically. The mass of the initial precursors was divided by the density of the material. In the case of MNPs, we using the mass fraction of MNPs in sol (2% wt.). The resulting whole volume of nanoparticles was divided by the average volume of single nanoparticle, assuming that nanoparticles were spherical. We added this explanation to Page 2-3, lines 93-96.

8) Figure 4B and Table 1S-6S: This referee would have expected that the sum of the intensities of the lower and upper bands is constant. This is not the case. Please comment.

It is expected that the sum of the intensities of the lower and upper bands would be consistent. In practice, this is not so due to differences in the stirring of test tubes, differences in pH of the sample solution and phoresies buffer, and pipette error. To reduce the impact of the above mention inaccuracies, we performed all experiments in 5 repetitions.

9) Aggregation of cationic particles could result from the addition of siRNA in the solution. Did the authors examine the colloidal stability of the NPs ?

Indeed, siRNA can cause aggregation of NPs. The initial stability of sols was very high due to high z-potential. Aggregation of NPs upon addition of siRNA was not additionally assessed because we were not intended to use these nanoparticles in any application. We observed no visible aggregation of NPs when mixed with siRNA. Furthermore, to release siRNA from nanoparticles, we adding PBS to the NPs+siRNA mixture, which in any case causing aggregation.

10) Bibliography should be improved (e.g., ref 11-12 are inadequate)

We have fixed references list typos.

11) Language editing is required.

            We have subjected the manuscript to another round of proofreading.

We hope that the revisions made will make our manuscript acceptable for publication in the Nanomaterials. Please do not hesitate to contact us if there are any questions regarding our revised submission.

Thank you for your cooperation.

            Sincerely,

Dr. Vladimir Vinogradov.  

Reviewer 2 Report

This manuscript by Prilepskii et al. describes the use of nanoparticles for increasing siRNA hybridization rate and found that cationic particles are more efficient. This is an interesting piece of work calling for the following remarks:

- As stated in the introduction (L30), "… a tempting question is the quality of the hybridization of double-stranded siRNA after interaction with inorganic nanoparticles." 
The authors do not really address this issue.

- Gold nanoparticles are initially referred to as GNPs. Then GNPs-15 and GNPs-30 appear, without explanation. Please comment.

- The size of NPs as determined by SEM, TEM, and DLS are not consistent (MNP : 7 to 60 nm ; SNP : 4 to 30 nm ; GNP : 30 to 60 nm). Please comment. Do the authors talk about nanoparticles or aggregates (as observed in Fig. 1 and fig. S1) ?

- As part of the discussion is focused on the interpretation of the zeta potential/size ratio, could the authors indicate/rationalize why they used the size as determined by DLS rather than by TEM or SEM?

- The nanoparticles described in the manuscript are likely titratable. The zeta value for such particles depends on pH. Thus, the pH value should be indicated in the text. Furthermore, the NP-RNA interaction is sensitive to pH. Consequently, this reviewer thinks that a proper comparison of the various NPs should be carried out in buffered solution, at a fixed pH.

- Complementary analytical data for citrate- and PEG-loaded MNPs should be provided to assess their decoration/coverage by citrate and PEG moieties (FT-IR, elemental analysis…). In addition, it is unclear how the various NPs were purified. A description should be provided.

- NP-assisted siRNA hybridization: Please indicate how was calculated/measured the number of NPs (6 X 108) in the samples.

- Figure 4B and Table 1S-6S: This referee would have expected that the sum of the intensities of the lower and upper bands is constant. This is not the case. Please comment.

- Aggregation of cationic particles could result from the addition of siRNA in the solution. Did the authors examine the colloidal stability of the NPs ?

Minor comments:

- Bibliography should be improved (e.g., ref 11-12 are inadequate)

- Language editing is required.

Author Response

Response to Reviewers remarks

We want to thank Reviewers for their time, and constructive suggestions are given for the manuscript entitled «Cationic Magnetite Nanoparticles for Increasing siRNA Hybridization Rates», written by Prilepskii et al.

We have carefully evaluated the comments and recommendations of the referees and revised the manuscript accordingly. Please find below a detailed point-by-point reply to all reviewer’s comments. We sincerely hope that we have succeeded in clarifying and answering all the queries that were raised.

Reviewer: 1

1) Figure 1 was referred to as “SEM images” in the text, where it should be “TEM images”.

We are sorry for this mistake; the corresponding change was made on Page 4, line 191.

2) The size distribution of MNP and SNP in Table 1 is different from the particles shown in Figure 1 (<10 nm particles). What is the potential reason for this difference? Please explain.

            The data presented in Table 1 measured via the DLS technique. Presented size is the hydrodynamical size and, in general, consists of the core (nanoparticle) and surrounding (solvent/water). In the case of small nanoparticles, hydrodynamical size can be 2-3 times bigger than the actual size of nanoparticle on SEM or TEM images. We added an explanation on Page 5, lines 199-204.

3) Given the different sizes of nanoparticles available, it would be interesting to see whether nanoparticle size (thus surface area) plays a role in the hybridization rate. Please discuss potential contribution of particle size or include some results on this direction.

            Indeed, it was our first thought that the bigger surface area of smaller nanoparticles (namely, magnetite) is beneficial for increasing of hybridization rate. However, gold nanoparticles with the same size according to TEM images (GNPs-15) and almost the same z-potential +35mV did not increase HR. Also, MNPsP differs from MNPs by only 10 nm in size, but this already leads to a decrease in HR. As we stated in conclusion, size certainly affects HR but only in conjunction with z-potential. We have added some additional discussion about this to the top of Page 9.

Reviewer: 2

1)  As stated in the introduction (L30), “… a tempting question is the quality of the hybridization of double-stranded siRNA after interaction with inorganic nanoparticles.” The authors do not really address this issue.

In this sentence, we wish to say about the amount of hybridized siRNA in comparison with non-hybridized. We changed the “quality” for “degree” to better represent our thoughts.

2) Gold nanoparticles are initially referred to as GNPs. Then GNPs-15 and GNPs-30 appear, without explanation. Please comment.

The number after abbreviation signifies the average diameter of the nanoparticles. We added an additional explanation of this on Page 2, Lines 86-87.

3) The size of NPs as determined by SEM, TEM, and DLS are not consistent (MNP : 7 to 60 nm ; SNP : 4 to 30 nm ; GNP : 30 to 60 nm). Please comment. Do the authors talk about nanoparticles or aggregates (as observed in Fig. 1 and fig. S1) ?

4) As part of the discussion is focused on the interpretation of the zeta potential/size ratio, could the authors indicate/rationalize why they used the size as determined by DLS rather than by TEM or SEM ?

We added an additional explanation on this issue on Page 5, lines 199-204. Briefly, we used hydrodynamic diameter from DLS because it is better representing the behavior of nanoparticles (especially of non-spherical forms) in media. Also, since ζ-potential measured by DLS too, we considered it more correct to calculate the “ζ-potential-to-hydrodynamic diameter ratio” instead of “ζ-potential-to-actual size”. As for the aggregates of SEM and TEM images, it originates from sample preparation. When a drop of sol is dried under vacuum before the electron microscopy observation, it is inevitably shrinking, causing aggregation of nanoparticles. Our observation has shown that all of the sols used were sedimentary and aggregately stable during the experiments. So, the presented in Figure S1 distribution is for separate nanoparticles.

4) The nanoparticles described in the manuscript are likely titratable. The zeta value for such particles depends on pH. Thus, the pH value should be indicated in the text. Furthermore, the NP-RNA interaction is sensitive to pH. Consequently, this reviewer thinks that proper comparison of the various NPs should be carried out in buffered solution, at a fixed pH.

We updated Section 2.5 on Page 3 with volume information of mixed reagents. For our experiments, we used very small amounts of NPs (0.5 to 1 µL, diluted ~100 times from stock solution) in comparison with buffered siRNA (20 µL). In this case, pH values for all samples were the same, around 5.5. Nevertheless, we added the pH values of NPs to Table 1 on Page 6.

5) Complementary analytical data for citrate- and PEG-loaded MNPs should be provided to assess their decoration/coverage by citrate and PEG moieties (FT-IR, elemental analysis…). In addition, it is unclear how the various NPs were purified. A description should be provided.

            All of the nanoparticles were purified by centrifugation according to original procedures described in references 21 and 22. We added this statement to Page 2, lines 92-93. We also added FTIR spectra for MNPs- and MNPsP to Supporting Information (Figures S3 and S4), and a short explanation to the manuscript on Page 6, as well as to Materials and Methods section on Page 3, lines 104-106.

6) NP-assisted siRNA hybridization : Please indicate how was calculated/measured the number of NPs (6 X 108) in the samples.

            Concentrations of nanoparticles (particles/mL) were calculated theoretically. The mass of the initial precursors was divided by the density of the material. In the case of MNPs, we using the mass fraction of MNPs in sol (2% wt.). The resulting whole volume of nanoparticles was divided by the average volume of single nanoparticle, assuming that nanoparticles were spherical. We added this explanation to Page 2-3, lines 93-96.

7) Figure 4B and Table 1S-6S: This referee would have expected that the sum of the intensities of the lower and upper bands is constant. This is not the case. Please comment.

It is expected that the sum of the intensities of the lower and upper bands would be consistent. In practice, this is not so due to differences in the stirring of test tubes, differences in pH of the sample solution and phoresies buffer, and pipette error. To reduce the impact of the above mention inaccuracies, we performed all experiments in 5 repetitions.

8) Aggregation of cationic particles could result from the addition of siRNA in the solution. Did the authors examine the colloidal stability of the NPs ?

Indeed, siRNA can cause aggregation of NPs. The initial stability of sols was very high due to high z-potential. Aggregation of NPs upon addition of siRNA was not additionally assessed because we were not intended to use these nanoparticles in any application. We observed no visible aggregation of NPs when mixed with siRNA. Furthermore, to release siRNA from nanoparticles, we adding PBS to the NPs+siRNA mixture, which in any case causing aggregation.

9) Bibliography should be improved (e.g., ref 11-12 are inadequate)

We have fixed references list typos.

10) Language editing is required.

            We have subjected the manuscript to another round of proofreading.

We hope that the revisions made will make our manuscript acceptable for publication in the Nanomaterials. Please do not hesitate to contact us if there are any questions regarding our revised submission.

Thank you for your cooperation.

            Sincerely,

Dr. Vladimir Vinogradov.  
